# From Reproduction to Replication: Evaluating Research Agents with Progressive Code Masking

**Gyeongwon James Kim, Alex Wilf, Louis-Philippe Morency, Daniel Fried**
Carnegie Mellon University
Pittsburgh, PA 15213, USA
`{gyeongwk,awilf,morency,dfried}@cs.cmu.edu`

## Abstract

Recent progress in autonomous code generation has fueled excitement around AI agents capable of accelerating scientific discovery by running experiments. However, there is currently no benchmark that evaluates whether such agents can implement scientific ideas when given varied amounts of code as a starting point, interpolating between *reproduction* (running code) and from-scratch *replication* (fully re-implementing and running code). We introduce AUTOEXPERIMENT, a benchmark that evaluates AI agents' ability to implement and run machine learning experiments based on natural language descriptions in research papers. In each task, agents are given a research paper, a codebase with key functions masked out, and a command to run the experiment. The goal is to generate the missing code, execute the experiment in a sandboxed environment, and reproduce the results. AUTOEXPERIMENT scales in difficulty by varying the number of missing functions $n$, ranging from partial reproduction to full replication. We evaluate state-of-the-art agents and find that performance degrades rapidly as $n$ increases. Agents that can dynamically interact with the environment (e.g., to debug their code) can outperform agents in fixed "agentless" harnesses, and there exists a significant gap between single-shot and multi-trial success rates (Pass@1 vs. Pass@5), motivating verifier approaches to our benchmark. Our findings highlight critical challenges in long-horizon code generation, context retrieval, and autonomous experiment execution, establishing AUTOEXPERIMENT as a new benchmark for evaluating progress in AI-driven scientific experimentation. Our data and code are open-sourced at https://github.com/j1mk1m/AutoExperiment.

## 1 Introduction

Advances in AI agents are reshaping how we approach scientific research (Lu et al., 2024), leading many to ask whether we have unlocked new forms of automated inquiry. However, we still lack benchmarks to evaluate the ability of AI agents to generate code for scientific experiments. While many benchmarks have been proposed to evaluate AI agents' capabilities on related code-generation tasks (Hendrycks et al., 2021; Chen et al., 2021; Yin et al., 2022; Jimenez et al., 2023), these have typically not evaluated agents' abilities to implement scientific code, which presents unique challenges. Recently, some benchmarks for scientific code generation have been proposed that require *reproducing* experimental results: confirming findings by running (but not implementing) the code (Bogin et al., 2024). Moreover, Starace et al. (2025) and Hua et al. (2025) proposed benchmarks on *replicating* results: confirming findings by writing code given only the description in the paper. Yet these benchmarks each have a fixed difficulty, and do not evaluate agents' abilities to implement experiments given the research paper and *varied portions of the original codebase*.

In this paper, we introduce the AUTOEXPERIMENT benchmark, which challenges agents to implement Python functions that have been removed from the original published research repositories, run the code in a sandboxed environment, and report the results of the experiment. As input, the agent is given the full research paper containing descriptions of the experiment, the command to run the experiment, and the codebase with a given number of target function(s) masked out. To evaluate

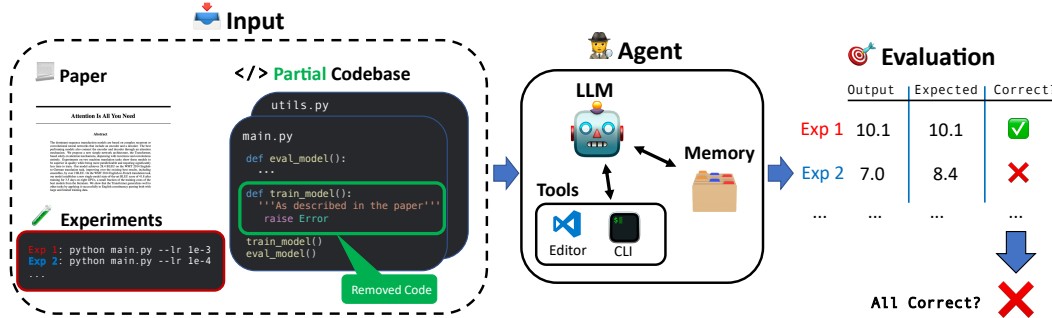

Figure 1: AUTOEXPERIMENT challenges AI Agents to implement and run experiments. The inputs are the papers with the description of the experiments (but not the results), the command sequences to run the experiments, and partial codebases where we mask out a *progressively increasing number* of functions from the original repository. The AI agent must implement the missing function(s), run the experiments using the command sequences, and correctly report the results of the experiments in order to pass the test cases.

correctness, the agent's findings are compared with the results of the same experiment run with the gold-standard published code from the paper. By progressively masking increasing amounts of the codebase, our benchmark can control the difficulty of the task, guiding a clear path from *reproduction* to *replication*. Our key findings are:

1. **Moving from reproduction to replication is hard, even for frontier models**: as we scale the difficulty of the task by *progressively masking* more of the code, we find that even the performance of agents powered by frontier language models falls off quickly, and that agents begin to rely increasingly on the natural language context from the research papers to bolster performance.

2. **Agent performance scales with dynamic interactivity and test-time compute**: agents that can *dynamically* make decisions on how to interact with the environment over time outperform agents in a *fixed* interaction harness. Agent performance also scales moderately with test-time-compute in reasoning models, although the trend is less pronounced.

3. **The Pass@k gap is large**: the pass rate improves over 10% from Pass@1 to Pass@5. This motivates future work on model-based verifiers for reranking, or agentive search approaches – which could recapture some of the performance of the pass@k scores.

## 2 THE AUTOEXPERIMENT BENCHMARK

AUTOEXPERIMENT evaluates agents' ability to write scientific code and run experiments. We collect four peer-reviewed research papers that have already been *replicated* as part of the ML Reproducibility Challenge (MLRC) (Rougier et al., 2017), mask out 85 functions from the publicly released codebases of those papers, and use the experiments and their results as "test cases" to evaluate the agent's implementation of the missing function(s). Papers from MLRC have been *replicated*, meaning that the method described in the paper has been reproduced and the results have been verified by another researcher. By limiting our selection to MLRC, our benchmark focuses on quality over quantity, guaranteeing that papers have a strong correspondence between natural language description and verified code implementation. We document the benchmark creation process in detail below.

**Collecting Research Papers**   Our key insight is that peer-replicated research papers can act as an effective testbed for creating agents capable of running experiments. The key requirement is that papers should be both *reproducible* (i.e., running the code with the command sequence for that experiment produces the corresponding result from the paper) and *replicable* (i.e., given only the contents of the paper, it is possible to write code from scratch to implement the experiments and arrive at the same results). To ensure *reproducibility*, we manually set up environments and run

the gold standard code with gold standard command sequences, verifying that all results from the research papers are reproduced. Furthermore, to ensure *replicability*, we select papers that have been replicated as part of the Machine Learning Reproducibility Challenge (MLRC) (Rougier et al., 2017). We manually performed additional filtering to ensure a high quality dataset: we select papers such that 1) the code repositories are well maintained with good documentation and (2) there is a strong mapping from research paper contents to the code. We list the details of the research papers in Appendix A.

**Masking Out Key Functions**  To construct each sample in the dataset, we collect Python functions that implement core logic for the experiments and replace their implementations with `NotImplementedError`. We select functions that produce errors when masked out, verifying that these functions are indeed essential in arriving at key experimental results. We experiment with masking out $n \geq 1$ functions at a time, treating each combination as a unique sample in the dataset. Our dataset comprises 85 unique functions, leading to 85 unique samples for $n = 1$, and up to $275,990$ possible samples for $n = 5$. To keep experimental runtime feasible, we select a maximum of 100 samples for evaluation for each setting of $n$. Although the number of samples is constant across $n$, as $n$ increases those samples are drawn from an increasingly diverse set of possible compositions.[1] These statistics are depicted in Table 1. The functions are on average 26.3 lines long and have an average of 15.9 function calls to standard library functions or other functions in the code.

**Evaluation Harness**  To evaluate the agent's output for each function, we collect a set of test cases: command sequences from the repository README that reproduce experimental results, and that rely on at least one of the masked out functions. We pass the command sequences to the agents as part of the input, and we ask the agent to output the results of the experiment in a structured format we can parse easily using a tool call described below. Our evaluation harness checks this output against the results from running the gold-standard code with the same command. We give agents access to the test command (instead of running the generated code in an evaluation harness offline) so that an agent can debug the code before submitting its final answer. To prevent cheating, we remove all numerical results from the research paper text.

We consider a sample to have passed our tests if the results for all tests differ relatively by no more than 5% from the gold standard results. The average number of testcases per sample across different values of $n$ is given in Table 1 below. Since the test cases are experiments that rely on any of the masked out functions, the average number of test cases increases monotonically with $n$.

| $n$ | 1 | 2 | 3 | 4 | 5 |
|---|---|---|---|---|---|
| Samples | 85 | 977 | 8046 | 52141 | 275990 |
| Test cases per Sample | 6.08 | 8.69 | 10.26 | 11.12 | 11.55 |

Table 1: Number of possible samples that can be generated given the codebase and average number of test cases per sample by $n$, where $n$ is the number of functions we mask out at once. Final evaluation is done over a fixed budget of a maximum of 100 samples for each setting of $n$.

In some cases, we take steps to shorten the runtime of the test cases – e.g., if a paper's experiment code involves training a model, reducing the number of training steps. In these cases, we ensure that our "curtailed" setting has high precision with the original test cases in its capacity to discriminate passing from failing solution attempts. See Appendix B for more details. On average, a complete agent run powered by GPT-4o took just over 11 minutes per sample, including both agentic actions and experiment runtime.

**Agent Architecture**  There are five key components that define each agent:

1. **Initial Prompt**: The system and user prompts describing the task to the agent. This prompt specifies the goal of the task, the function that has been masked out, and the command to run the experiments. The exact prompt we used can be found in Appendix J.1.

---

[1]The number of samples we have for different values of $n$ increases combinatorially for each paper. E.g. a paper with 23 functions will contribute $\binom{23}{n}$ samples for each $n$. Note, the increase happens individually for each paper, so the number of samples do not increase combinatorially in total.

2. **Tool Definitions**: The exact tool definitions we use are similar to the ones found in Huang et al. (2024), which give the agents the ability to navigate the repository, manipulate files, and execute scripts. A high level description of these tools can be found in Table 2.

3. **Step-by-Step Prompting Strategy**: At each step, the agent can be prompted to reason and output actions in different ways. We experiment with different popular strategies (Yao et al., 2023; Song et al., 2023; Huang et al., 2024), described in Appendix J.2. For the main experiments, we use the best performing strategy, ReAct (Yao et al., 2023). This is a light-weight strategy, which prompts for a natural language thought and action at each step, allowing the agent to reason about previous action history and determine the best next action.

4. **History Management**: As the agent continues to interact with the environment, the history of its interactions grows. How agents manage that history is the subject of recent research papers (Wang et al., 2024b). We experiment with different common approaches, described in Section C. For the main experiments, we use Full history management, which performed best when combined with ReAct prompting. This strategy simply maintains the full list of previous actions and corresponding environment observations, which is incorporated into the prompt at each step.

5. **Backbone LLM**: We evaluate different underlying LLMs: GPT-4o, GPT-4o-mini, Claude-3.5-Sonnet, and Claude-3.7-Sonnet. We also tested open-source models Qwen2.5-Coder-32B-Instruct and All Hands' openhands-lm-32b-v0.1, but found that these models struggle to make any progress due to insufficient formatting, tool-calling, and instruction-following skills.

| Action | Input | Observation |
|---|---|---|
| Execute Command | Command | Executes shell command |
| List Files | Directory | Lists all files and folders in directory |
| Execute Script | File name, arguments | Output of executing script |
| Move | Source, destination | Moves file or dir to destination |
| Edit File | File name, edit instruction, save file | Edit based on instruction (LLM call) |
| Write file | File name, content | Writes content to a file |
| Inspect File Lines | File name, start line no., end line no. | File content between line numbers |
| Understand File | File name, things to look for | Relevant information (LLM call) |
| Change Directory | Directory | Changes the working directory |
| Final Answer | Final answer | Submits the answer in JSON format |

Table 2: Tools available to the agent. A complete description of these tools in JSON format can be found in Appendix J.3.

**Environment and Resources** To ensure safety and guarantee reproducibility, we create a sandboxed environment using Docker where the agent can edit the code and run command line instructions within conda environments already set up to run code for each repository. We run all experiments on a Linux machine with AMD 24-Core processor and 2x NVIDIA GeForce RTX 3090 Ti GPUs. To limit runtime for the exponentially increasing number of samples, we evaluate on only 100 samples for higher values of $n$, described in Appendix E. For each sample, agents were limited to 50 action-taking steps to limit the resources used by each agent, a maximum runtime of 30 minutes, and maximum LLM compute budget of $1. The average compute time and cost of each run are detailed in Appendix K. Appendix L contains an example of a full agent trajectory.

**Data Contamination** Recent LLM benchmarks have shown to be susceptible to data contamination (Xu et al., 2024). With the release of new models, it is crucial to have contamination-free benchmarks that can fairly evaluate the models' capabilities. We evaluate the data contamination risk for AUTOEXPERIMENT. First, the agent performance on our benchmark at the easiest setting ($n = 1$)

is relatively low ($< 40\%$), suggesting that contamination is unlikely. We also performed a data contamination test (Shi et al., 2024) on models: Qwen2.5-1.5B-Instruct, Qwen2.5-Coder-32B-Instruct, and All Hands' openhands-lm-32b-v0.1. The results show that on AutoExperiment, models achieve a contamination score of 35%, 51%, and 57%, respectively, which are all far below the threshold to be considered contaminated (85% according to developers of Shi et al. (2024)). Moreover, if we detect data contamination in the future, a new version of the dataset can be created according to the detailed process outlined in our paper. As AI research continues to grow (Azad & Banu, 2024), there will be an increasing source of high-quality research papers with code repositories that can be used to create the dataset.

## 3 EXPERIMENTS

### 3.1 FROM REPRODUCTION TO REPLICATION: REMOVING $n \geq 1$ FUNCTIONS

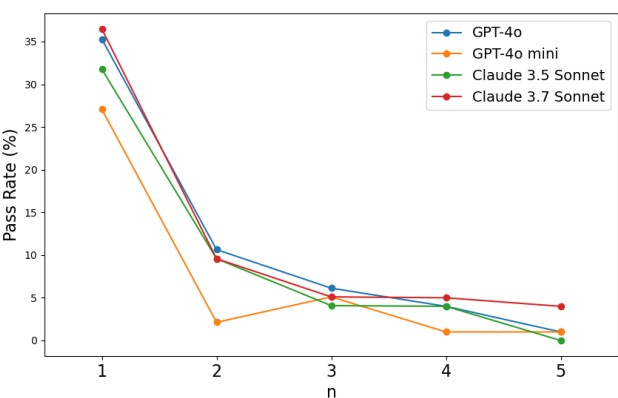

Figure 2: Performance on AUTOEXPERIMENT with $n$ functions removed at a time.

We experimented with masking out multiple functions at once from the codebase, using agents powered by different frontier models. Agents perform reasonably well with 1 function removed. Claude-3.7-sonnet leads with 36.5% pass rate followed by GPT-4o with 35.3%, Claude-3.5-sonnet with 31.8%, and GPT-4o-mini with 27.1%. Interestingly, performance plummets as soon as $n = 2$ with Claude-3.7-sonnet and Claude-3.5-sonnet having 9.6% pass rate followed by 8.5% for GPT-4o and 2.1% for GPT-4o-mini – a performance drop of 70-90% on average. As $n$ increases, performance continues to drop with most models reaching negligible pass rates by $n = 5$.

This precipitous drop is an important result because it shows that although frontier models perform well with one function removed, the benchmark becomes significantly more difficult as it nears the "from-scratch replication" setting, with all functions removed. Because we can control the number of functions removed from the gold codebase, we control a "knob" of difficulty for the benchmark, guiding the path towards full replication.

### 3.2 THE PASS@K GAP

One important metric in code generation benchmarks is **Pass@k**, in which the model generates $k$ complete solutions to each problem and we pick the best one (as defined by the ground-truth test cases). This is important because it can act as an important indicator for the potential of **search and verification** strategies: a simple search strategy (e.g., generate 5 solution attempts) combined with a perfect verifier yields the Pass@5 performance.

We examined the Pass@k results for GPT-4o and Claude-3.5 Sonnet and found a large gap between Pass@1 and Pass@5 for our benchmark: $35.3\% \rightarrow 48.2\%$ for GPT-4o and $31.8\% \rightarrow 42.2\%$ for Claude 3.5 Sonnet. This offers substantial potential that there could be large gains in performance on this benchmark from **better verifiers alone**. In addition, this may motivate research into improving the policy function through **RL**.

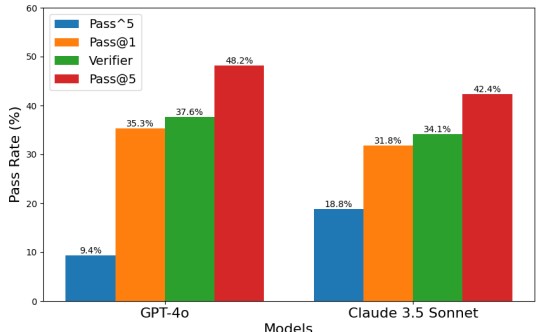

Figure 3: We test agents with GPT-4o and Claude 3.5 Sonnet on $n = 1$ setting. The gap between Pass@1 (orange) and Pass@5 (red) performance is large. This is promising for lines of research into **verification** for use as part of search. We also examine "Verifier", which uses the model as its own verifier to select the best answer before submitting that for evaluation. Models-as-verifiers can help somewhat on this task, but there is still a substantial gap to oracle-level verification. We also report Pass^5 (all 5 responses are correct).

We also investigated using the model itself as a verifier ("Verifier"), to see **how much of the Pass@5 gap we can capture through current model verification capabilities** (and how much room there is to improve models as verifiers for this task). We found that models used as their own verifiers could improve performance somewhat but there is still a significant gap to oracle-level performance as shown in Figure 3. Appendix F describes the prompt used for the model verifier in more detail.

Lastly, we looked at Pass^k, which measures whether all $k$ responses are correct. Overall, the differences between Pass^5, Pass@1, and Pass@5 metrics are much higher for GPT-4o than Claude-3.5-Sonnet, suggesting that GPT-4o has more variance in its responses and chosen actions.

### 3.3 SCALING INTERACTIVITY AND TEST-TIME COMPUTE IN AGENTS

Recently, o1 was able to achieve state-of-the-art results on SWE-Bench, a benchmark designed to benchmark *agents*, by using an "agentless" harness in which a fixed set of steps was applied, using an LLM to perform the individual steps (localization, repair, and patch validation) (Xia et al., 2024). Motivated by this work, we investigate whether it is helpful for the agent to have the ability to **dynamically** decide which sequential interactions to take with the environment (our main setup for all other experiments), or whether our task can be solved using a **fixed** set of steps with a strong model applied to each step. We run these experiments with the o3-mini model because using the full o1 for each step is very costly in the dynamic setting. We adapted the agentless harness to our task, where the pipeline consists of text and code retrieval steps, followed by code-infilling, and lastly, running the experiment and extracting results. In the retrieval steps, we use an embedding model (OpenAI text-embedding-3-small) to embed chunks of text (from the paper) and code (from the repository), respectively, and retrieve top-$k$ relevant context using cosine similarity. This retrieved context is included in the prompt that tasks an LLM to infill the missing function. Lastly, we run the experiment using the given command sequence and extract the results from the execution outputs with an LLM. The exact prompts and experimental cost are detailed in Appendix H.

| Interaction Type | GPT-4o | o3-mini |
|---|---|---|
| **Fixed** | 8.3 | 27.8 |
| **Dynamic** | **35.3** | **33.3** |

Table 3: Agents powered by GPT-4o and o3-mini backbones benefit from the capability of dynamically choosing which action to take next in interactions with the environment, as opposed to the fixed three step "agentless" harness.

We found that **dynamically-interactive** agents far outperformed **fixed-interaction** agents across both reasoning- (o3-mini) and non-reasoning models (GPT-4o). With a fixed interaction harness, a GPT-4o powered agent achieved a pass rate of only 8.3%. When given access to dynamic interaction capability, that jumped to 35.3%, a more than **4.0x** improvement. Performance gains for o3-mini were smaller, yet still substantial: 27.8% to 33.3%. This highlights the importance of agents being able to dynamically interact with the environment. There may exist a hand-crafted agentless-style fixed interaction paradigm that optimizes for this benchmark, but from our investigations it seems that this would be less fruitful than allowing agents to dynamically choose how to interact with their environment.

Part of the explanation for this performance improvement may be due to the effectiveness of debugging which can be performed by the dynamic agents. Often, the first attempt crashes (69.4% of the time for agents): it results in non-working code that produces a runtime or compilation error. Agents can consequently debug and produce working code. We find that 29.1% of runs recover to a working state and 18.6% of runs end up with the correct working code. However, reasoning-only models operating in the fixed interaction paradigm – without access to multiple debugging steps – were not able to achieve these gains. o1's crash rate was 75.0% and o3-mini's was 66.7%. These results suggest that debugging may partially explain why the dynamic agent outperforms the fixed decomposition setup on this benchmark. Appendix D provides more detail and examples of common errors that agents face.

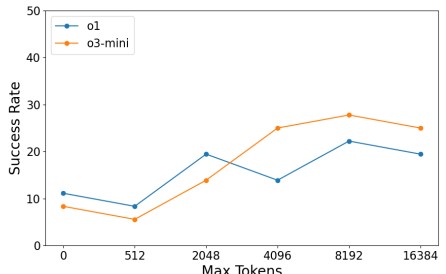

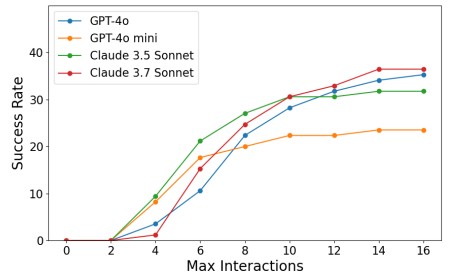

(a) Performance scaling across **Max Reasoning Tokens** in the **Fixed** setting.

(b) Performance scaling across **Max Interactions** in the **Dynamic** setting.

Figure 4: Scaling the number of reasoning tokens in the fixed-interaction setting (left) does not scale well, in comparison to scaling the number of interactions in the dynamic setting (right).

Another recent interesting result is that reasoning models (OpenAI, 2024; DeepSeek-AI et al., 2025) can improve performance with "test-time compute" by, e.g., increasing the max number of tokens they use in their chain-of-thought before probing for a final answer (Muennighoff et al., 2025). We also investigate whether scaling test-time compute helps on our task. We test this in the fixed-interaction setting, because with only one action taken by the agent, we can easily control the number of reasoning tokens used (as opposed to constraining per-action). We reproduce the experimental method from Muennighoff et al. (2025), in which we constrain the model's reasoning tokens to some fixed number, then ask the model to write code given the reasoning trace. The exact prompts and experimental details are in Appendix I.

The results of our experiment are depicted in Figure 4. Similarly to other reasoning benchmarks (Muennighoff et al., 2025), scaling max tokens in the fixed-interaction setting improved performance, but the increase was moderate. Running the harness with 0 reasoning tokens achieves 11.1% and 8.3% for o1 and o3-mini respectively, and the performance reaches its peak at 22.2% and 27.8% for o1 and o3-mini before tapering off. Although performance does improve with reasoning tokens, there seems to be an upper limit on the performance. On the other hand, scaling the max interactions allowed by the agents significantly outperformed the Fixed setting with performance saturating around 38.9%.

The key takeaways from these experiments are that (1) AUTOEXPERIMENT provides a testbed for **dynamically interactive agents**, that isn't easily solvable by a fixed, agentless interaction paradigm. (2) while scaling max reasoning tokens improves performance moderately, **scaling the number of**

**steps of agentic interactions** has a much more substantial effect. We believe these experiments highlight the strength of our benchmark in evaluating interactive research agents.

### 3.4 DEPENDENCE ON NATURAL LANGUAGE

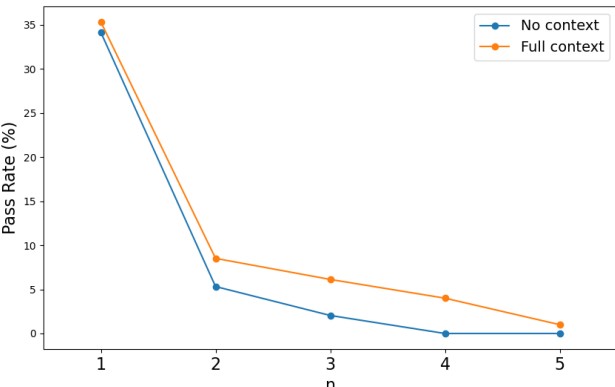

Figure 5: Performance across different values of $n$, under different natural language (the text of the paper) retrieval conditions. **No** retrieval only gives agents access to the code. **Full** retrieval additionally gives agents access to the paper through tool calls described in Appendix J.3 (our default setting for all other experiments). We test using agents with GPT-4o as backbone, ReAct prompting, and Full history management. We find that access to the paper becomes **more important as more functions are masked out**.

Another key finding is the relationship between $n$ and the agent's access to natural language context (relevant text from the paper that guides code generation). To understand the importance of natural language context, we run agents on the **No** context setting, which removes the research paper contents from the agents' inputs. On the contrary, our main setting (**Full** context) provides the contents of the paper as a text file that agents can read from. We found that models were often able to perform fairly well on the $n = 1$ task (where only a single function needs to be implemented) without access to the natural language (the **No** context setting). However, as $n$ increases, performance degrades more rapidly in the **No** context setting, suggesting that natural language guidance becomes more important as greater proportions of the codebase is masked out.

At $n = 1$, **No** and **Full** context perform similarly with $35.3\%$ pass rate for **Full** context and $34.1\%$ for **No** context. This is somewhat surprising since we expect the implementation details to come from the paper. In qualitative error analysis, we found that this was because information required to implement some of the functions was often inferrable from other functions in the codebase. Also, the relatively low performance of agents in the **Full** context setting could be explained by the often-observed phenomenon that language models are distracted by irrelevant context (Creswell et al., 2022). Yet as $n$ increases, natural language context from the paper become more important. We find that performance drops more rapidly in the **No** context setting such that by $n = 4$, performance reaches complete failure. We also manually defined "Oracle" retrievals for relevant passages for each function, but did not see a significant improvement to using this over the full context of the paper. Qualitatively, we found that the agents often focused on mimicking the style of the code around the target function(s) more than the natural language from the paper. As such, we found that having the right code context is important. We show results on a similar experiment with code context in Appendix G.

## 4 RELATED WORK

**Automating Scientific Discovery** It has been a long-standing dream in AI research to use machines as tools for new scientific discovery (Simon, 1977; Langley, 2000). Recent advancements in generative AI agents have led to progress on important capabilities such as idea generation (Qi

et al., 2023; Wang et al., 2024a), peer review (Tyser et al., 2024; Liu & Shah, 2023), and writing aids (Altmäe et al., 2023; Salvagno et al., 2023). As AI grows more capable of running experiments as well (Lu et al., 2024) effective benchmarking of agents' abilities to run experiments becomes all the more important.

**Generative Agents**   Generative Agents, agents powered by large generative models with access to interactive environments (Xi et al., 2023), have shown promising results on a variety of tasks (Ahn et al., 2022; Huang et al., 2023). Some core challenges of generative agents are inherited from their generative backbones, including long context retrieval, reasoning, planning, and self-reflection (Shinn et al., 2023; Madaan et al., 2023). Other core challenges are novel to the agentic setting, including tool use to interact with their environments and managing long histories of interactions (Wang et al., 2024c; Zhang et al., 2024).

**Benchmarking Code Generation**   Code-generation benchmarks have traditionally evaluated systems' ability to write code from a docstring, evaluated by unit tests (Hendrycks et al., 2021; Chen et al., 2021). This paradigm has been extended to data science notebooks  (Yin et al., 2022) as well as to patch generation for GitHub issues (Jimenez et al., 2023). Most similar to our work are benchmarks focused on testing agents' ability to perform ML research. MLAgentBench (Huang et al., 2024) focuses on agents' abilities to improve model performance, but does not test on other research hypotheses. Similarly, MLGym (Nathani et al., 2025) tests agents on AI research tasks involving data processing and model training for better performance. SUPER (Bogin et al., 2024) tests agents on *reproducing* results given the full repository (generating commands, not code), and PaperBench (Starace et al., 2025) tests agents on *replicating* results from scratch. Concurrent to our work is ResearchCodeBench (Hua et al., 2025), which tests core code implementations from recent research papers. However, none have tested agents' abilities to implement research ideas by masking out *varying amounts* of the gold standard code. As we have shown, this leads to an important knob with which to control the difficulty of the benchmark and the reliance of the model on natural language vs. code context.

## 5   CONCLUSION

We introduced AUTOEXPERIMENT  a benchmark designed to evaluate AI agents on the task of implementing and running scientific experiments from natural language descriptions in research papers. By varying the proportion of functions removed from research repositories, AUTOEXPERIMENT bridges the gap between reproduction and full replication, offering a testbed for autonomous research agents. We demonstrate that agent performance degrades sharply with increased autonomy, that dynamic decision making is critical (e.g., for debugging), and that multi-trial accuracy outpaces single trial significantly, highlighting importance of verifiers. AUTOEXPERIMENT offers a critical lens on the capabilities and limitations of current frontier models, and provides a foundation for developing agents capable of autonomous scientific discovery.

## LIMITATIONS

Our benchmark focuses exclusively on machine learning papers and Python-based code. It is unclear how generalizable our findings are to other scientific domains and programming languages. The dataset is relatively small (85 tasks at $n$=1) and resource-intensive to run, though parallelization is possible. All experiments are executed in Conda environments; while agents may attempt package installation, this shifts the task definition slightly toward open-ended setups such as SUPER (Bogin et al., 2024)—though we did not observe this behavior in practice.

## REPRODUCIBILITY STATEMENT

To ensure reproducibility, our code and data is open-sourced at this link. We share details about our development environment in Section 2. Furthermore, all prompts used in this paper are shared in Appendices F, H, and J.

ACKNOWLEDGEMENTS

We thank Alex Tianyi Xu for building the compute infrastructure used in this project. We are grateful to Yiqing Xie and Saujas Vaduguru for their valuable feedback, and to Alex Hsia for assistance with the initial data collection. This material is based upon work partially supported by the National Institutes of Health (awards R01MH125740, R01MH132225, and U01MH136535). Any opinions, findings, conclusions, or recommendations expressed in this material are those of the author(s) and do not necessarily reflect the views of the sponsors, and no official endorsement should be inferred.

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

## A  PAPERS

| Papers | Repositories | Functions |
|---|---|---|
| ITI-GEN: Inclusive Text-to-Image Generation | Link | 23 |
| Studying How to Efficiently and Effectively Guide Models with Explanations | Link | 33 |
| Cartoon Explanations of Image Classifiers | Link | 14 |
| Joint Multisided Exposure Fairness for Recommendation | Link | 15 |

Table 4: Papers in AUTOEXPERIMENT.

All papers were submitted to the ML Reproducibility challenge, which require papers and their associated code repositories to be licensed under Creative Commons 4.0 License. This allows for copying, modifying, and sharing of the original work.

## B  CURTAILED TEST CASES

| Metric | Recall | Precision |
|---|---|---|
| Unweighted | 0.949 | 0.959 |
| Weighted | 0.922 | 0.959 |

Table 5: Precision and Recall of using curtailed testcases as success metric with respect to full testcases.

We run the $n = 1$ setting with GPT-4o with the curtailed experiments and the full experiments and observe how effective the curtailed experiments are as a proxy for the test case defined by the full experiments. On a run with 92, 49 correct and 43 incorrect, we compared the decisions from the curtailed and full experiments, and find that both the precision and recall are very high, meaning that the curtailed testcases are an effective proxy for the full experiments. The curtailed experiments took significantly less time and compute resources to run, making the benchmark feasible for large scale testing.

## C  AGENT ARCHITECTURES & BACKBONE MODELS

| | GPT-4o | | | | GPT-4o-mini | | |
|---|---|---|---|---|---|---|---|
| | Full | Sm (5) | Sl (5) | Sl (1) | Full | Sm (5) | Sl (5) | Sl (1) |
| **ReAct** | **38.9** | 27.0 | 32.4 | 27.0 | 24.3 | **29.7** | 21.6 | 18.9 |
| **MLAgentBench** | 29.7 | 29.7 | 24.3 | 32.4 | 8.1 | 21.6 | 13.5 | 24.3 |
| **Planning-Only** | 18.9 | 18.9 | 18.9 | 16.2 | 18.9 | 24.3 | 21.6 | 27.0 |

Table 6: Success Rates for different prompting strategies and history management combinations.

Many works (Yao et al., 2023; Song et al., 2023; Huang et al., 2024; Yang et al., 2024) have investigated prompting frameworks for agents designed to elicit planning, action, and reflection capabilities. These frameworks are often tightly coupled with how to manage the history of interactions that gets passed to the agent's context window, especially as the number of interaction steps with the environment grows. The **prompting frameworks** we investigate are

1. **ReAct** (Yao et al., 2023): prompts for a natural language thought and the next action.
2. **MLAgentBench** (Huang et al., 2024): prompts for a highly structured response including a thought, research plan, and reflection in addition to the next action.

3. **Planning-Only** (Song et al., 2023): prompts for a step-by-step plan on how to achieve the final goal and the current status of this plan in addition to the next action.

We investigate these across different approaches to **managing the history** of interactions, including

1. **Full**, which passes the entire history of interactions,

2. **Sliding Window – Sl (k)** which passes only the most recent $k$ interactions

3. **Summary – Sm (k)**, which passes the most recent $k$ interactions as well as a summary of older interactions.

We find that the choice of prompting technique and memory management strategy impacts agent performance. Among the evaluated methods, ReAct outperforms both Planning and MLAgentBench across most configurations, particularly with the GPT-4o model, achieving a success rate of **38.9%** with full memory. Notably, full memory management yields the best results with ReAct, though this advantage is contingent on the prompting approach and the model's capacity for handling long context (e.g., GPT-4o-mini struggles with longer context and benefits from summarization instead of full context with ReAct). For Planning prompts, which typically require longer context windows, summarization strategies (Sm and Sl) are critical—especially for smaller models like GPT-4o-mini. Overall, it seems that **jointly optimizing prompting and memory frameworks** may be important to maximize performance given a model's capacity for long-context reasoning. We use the paired bootstrap test (Efron & Tibshirani, 1994) to compare the performance of the best agent combination (ReAct + Full + GPT-4o) with the worst (MLAgentBench + Full + GPT-4o mini) and find statistically significant difference (p-value 0.004). However, the best agent compared with the 2nd best (React + Sl(5) + GPT-4o) yields p-value of 0.243. This warrants further investigation into optimizing agent architectures, which we leave for future work.

## D    ERROR ANALYSIS

We analyzed agent trajectories to understand the common failure modes on AutoExperiment. For our main agent (ReAct prompting and full memory with GPT-4o backbone on n=1 setting), 35.3% of runs resulted in the correct answer, 31.4% produced an incorrect answer, and 33.3% did not produce an answer at all.

For instances in which the agent produced an incorrect answer, we found that the agent wrote code that compiled and ran without errors, but had implementation flaws that do not reflect the natural language description in the paper. For instance, one paper describes a novel metric function with a precise formula, which is to be implemented in the missing function. The agent often fails to retrieve this information from the full paper text and relies solely on code context to write the missing function. This leads to the implementation having operations that do not match the correct formula.

In the runs that failed to produce an answer, agents failed to debug their code as agents often produced code that was incompatible with the rest of the code base. For instance, agents would use variables or call functions that are not defined, or misunderstand the type of variables.

## E    EVALUATION SAMPLES FOR HIGHER VALUES OF $n$

| $n$ | 2 | 3 | 4 | 5 |
|---|---|---|---|---|
| Number of samples | 94 | 98 | 100 | 100 |

Table 7: Number of samples used for evaluation for $n \geq 2$.

Because the number of possible samples increases exponentially for higher values of $n$, we test on a subset for each $n$. The samples were chosen at random, and the number of samples for each $n$ are listed above.

## F  VERIFIER PROMPT

> You are an expert software engineer. You are given various code snippets that are generated by an LLM that implement function {function_anem} according to { function_docstring }. Your job is to verify the correctness of the code and pick the best candidate. Here are the code snippets:
>
> Code 1:
> {contents of code 1}
>
> Code 2:
> {contents of code 2}
>
>  ...
>
> Code 5:
> {contents of code 5}
>
> Please pick the best candidate and return the name of the code snippet. Explain your reasoning.

## G  CODE CONTEXT RETRIEVAL RESULTS

| No | Full | AST | Embedding |
|------|------|------|------|
| 13.9 | 36.1 | 33.3 | **41.7** |

Table 8: Pass rate of agents with No retrieval, Full (All code in context window), AST-based, and Embedding-based code retrieval

We observe that embedding-based retrieval significantly outperforms both in-context (Agent-led) retrieval and other retrieval strategies (Table 8). Agents using embedding-based retrieval achieve a pass rate of 41.7%, compared to 36.1% with in-context retrieval and 33.3% with AST-based retrieval. Simply placing all relevant code into the context window degrades performance, underscoring the limitations of context window-based reasoning. This result highlights code retrieval as a central challenge in our benchmark and a key area for future research.

## H  AGENT-LESS EXPERIMENT

We follow the Agentless approach for SWE-Bench (Xia et al., 2024), which consists of localization, repair, and patch validation steps. One key difference is that our task does not require localization since the location of the missing function is given. However, in order to write the masked-out function, retrieval of relevant natural language context (from research paper) and code context (rest of repository) is necessary. As such, our "agentless" harness consists of the following steps:

1. **Natural language retrieval**: we divide the research paper contents into paragraphs and use an embedding-based retrieval method using the function header and docstring as query to retrieve top-$k$ relevant text snippets. We use OpenAI's text-embedding-3-small model to compute embeddings.

2. **Code retrieval**: LLM-based prompting method is used to narrow down on the relevant files in the repository. Then, we embed each code snippet and use function header and docstring as query to retrieve top-$k$ relevant code snippets. We use OpenAI's text-embedding-3-small model to compute embeddings.

3. **Code infilling**: this step is performed by prompting the LLM to fill in the masked-out function using natural language and code context from the retrieval steps. We reproduce

the approach from Muennighoff et al. (2025), in which we first prompt for reasoning, then prompt for the code implementation. More details of how we can control the number of reasoning tokens are in Appendix I.

4. **Validation and Extraction**: we evaluate the generated code using the provided experiments (testcases) and extracts the final results.

## H.1 CODE RETRIEVAL FILE LOCALIZATION PROMPT

```
Given repository  structure , return  top 10 file  paths  related  to the  code.
Repository  structure :
{ repository_structure }

Code:
{ function_header_and_docstring }

Return  only  a  new line−separated  list  of paths  like  this :
```
./ relative /path/to/ file1
./ relative /path/to/ file2
...
./ relative /path/to/ file10
```
```

## H.2 CODE INFILLING PROMPT

**Reasoning prompt**

```
You are  a  helpful  coding  assistant . You are  given  contents  of a Python  file  with
     one missing  function . Paper  context  contains  snippets  of a research  paper
     that  describes  how to implement the code. Code context  contains  code  snippets
      that  are  similar  to the  missing  function .

### Paper  Context  ###
{ paper_context }

### Code Context  ###
{ code_context }

### File  Content  ###
``` python
{ file_content }
```

Think  about  how you want to  implement the  missing  Python  function .
```

**Completion prompt**

```
### Thought ###
{thought}

### File  Content  ###
``` python
{ file_content }
```

### Python  function  ###
``` python
{ function_content }
```

```
‘‘‘

Give only the missing function implementation. Provide only the code in markdown
        format. I.e. ‘‘‘ python
```

## I  TEST TIME COMPUTE EXPERIMENTS

We reproduce the experimental setting from Muennighoff et al. (2025), in which we can control the number of reasoning tokens. In the code infilling step (see Appendix H), we first prompt for reasoning followed by another prompt to perform the code infilling. When the reasoning step prematurely terminates before reaching the desired token count, we append the "Wait" token to the response, forcing the model to reason for a controlled number of tokens. In this experiment, we report the success rate of the agentless harness with o1 and o3-mini as model backbones and maximum tokens ranging from $[0, 16384]$.

| Reasoning Models | 0 | 512 | 2048 | 4096 | 8192 | 16384 |
|---|---|---|---|---|---|---|
| **o1** | 11.1 | 8.3 | 19.4 | 13.8 | 22.2 | 19.4 |
| **o3-mini** | 8.3 | 5.6 | 13.8 | 25.0 | 27.8 | 25.0 |

Table 9: Scaling number of reasoning tokens in the **Fixed** setting.

## J  AGENT ARCHITECTURE DETAILS

### J.1  INITIAL PROMPT

```
### Setting
You are a research  assistant  that is  tasked  with  running  experiments  to  produce
        results  for  a  scientific  paper.
The directory  already  contains  some code that  implements  the  experiments  done in
        the  paper  and  the  environment  is  already  set  up. But the  implementation  is
        incomplete  in  that  there  are  functions  not  implemented yet.

You can use the  following  tools  to  interact  with  the  environment.
### Tools
{ tools }

Your task is  to  write  the  missing  functions  in  the  code and  running  ‘bash  refsol .
        sh‘ to  obtain  experiment  results .
Here are the  experiments  you need to  report :
{experiment  details }
```

### J.2  PROMPTING STRATEGIES

**ReAct**

```
Think about  what  action  to  perform  next .
```

**MLAgentBench**

```
Always respond in  this  format  exactly :
 Reflection : What does the  observation  mean? If  there  is  an  error , what caused the
        error  and how to debug?
```

> Research Plan and Status: The full high level research plan, with current status and confirmed results of each step briefly annotated. It must only include progress that has been made by previous steps. If there is any update, enclose the new update text in double asterisks **like this**. If there is no update, just copy the previous step Research Plan and Status. The high level plan from the previous step should be fully retained, unless it is intentionally revised.
>
> Fact Check: List all objective statements in the updates to Research Plan and Status one by one and point out whether it is guessed versus directly confirmed by the previous observation directly above. Performance numbers can only be confirmed by running the code and observing the output.
>
> Thought: What you are currently doing, what actions to perform and why

**Planning-Only**

> Create a high level plan with current status and confirmed results.

## J.3 TOOL CALLS

We follow a standard tool set for coding agents inspired from MLAgentBench (Huang et al., 2024) and SWE-Agent (Yang et al., 2024). In summary, the agent can navigate the repository, read-/write/edit files, and execute Python or bash scripts.

```
tool_prompt = [
    {
        "type": "function",
        "function": {
          "name": "final_answer",
          "description": "Use this to submit the final answer to the
              ↪ current task",
          "required": ["final_answer"],
          "parameters": {
            "type": "object",
            "properties": {
              "final_answer": {
                  "type": "string",
                  "description": "json format string representing
                      ↪ dictionary of the final answer"
              }
            },
          }
        }
    },
    {
        "type": "function",
        "function": {
          "name": "understand_file",
          "description": "Use this to read the whole file and
              ↪ understand certain aspects. You can provide a
              ↪ detailed description on what to look for and what
              ↪ should be returned.",
          "required": ["file_name"],
          "parameters": {
            "type": "object",
            "properties": {
              "file_name": {
                  "type": "string",
                  "description": "a valid file name with relative
                      ↪ path to current directory if needed",
              },
              "things_to_look_for": {
                  "tupe": "string",
```

```
                            "description": "a detailed description on what to
                                ↪ look for and what should returned"
                        }
                    }
                },
            }
        },
        {
            "type": "function",
            "function": {
                "name": "inspect_file_lines",
                "description": "Use this to inspect specific part of a file
                    ↪ precisely, or the full content for short files.",
                "required": ["file_name"],
                "parameters": {
                    "type": "object",
                    "properties": {
                        "file_name": {
                            "type": "string",
                            "description": "a valid python script name with
                                ↪ relative path to current directory if
                                ↪ needed"
                        },
                        "start_line_number": {
                            "type": "number",
                            "description": "a valid line number"
                        },
                        "end_line_number": {
                            "type": "number",
                            "description": "a valid line number"
                        }
                    }
                },
            }
        },
        {
            "type": "function",
            "function": {
                "name": "edit_file",
                "description": "Use this to do a relatively large but
                    ↪ cohesive edit over a python script. Instead of
                    ↪ editing the script directly, you should describe the
                    ↪  edit instruction so that another AI can help you do
                    ↪  this.",
                "required": ["file_name", "edit_instructions"],
                "parameters": {
                    "type": "object",
                    "properties": {
                        "file_name": {
                            "type": "string",
                            "description": "a valid file name with relative
                                ↪ path to current directory if needed. An
                                ↪ empty file will be created if it does not
                                ↪ exist."
                        },
                        "edit_instruction": {
                            "type": "string",
                            "description": "a detailed step by step
                                ↪ description on how to edit it."
                        },
                        "save_name": {
                            "type": "string",
                            "description": "a valid file name with relative
                                ↪ path to current directory if needed"
                        }
```

```
                }
            },
        }
    },
    {
        "type": "function",
        "function": {
            "name": "write_file",
            "description": "Use this to write content to a file. If the
                ↪  file does not exist, a new file will be created. If
                ↪  file exists, content will be overriden",
            "required": ["file_name", "content"],
            "parameters": {
                "type": "object",
                "properties": {
                    "file_name": {
                        "type": "string",
                        "description": "a valid file name with relative
                            ↪ path to current directory if needed"
                    },
                    "content": {
                        "type": "string",
                        "description": "the content to be written to the
                            ↪ file"
                    }
                }
            }
        }
    },
    {
        "type": "function",
        "function": {
            "name": "execute_python_script",
            "description": "Use this to execute the python script. The
                ↪ script must already exist.",
            "required": ["file_name"],
            "parameters": {
                "type": "object",
                "properties": {
                    "file_name": {
                        "type": "string",
                        "description": "a valid python script name with
                            ↪ relative path to current directory if
                            ↪ needed"
                    },
                    "arguments": {
                        "type": "string",
                        "description": "command line arguments to use if
                            ↪ needed"
                    }
                }
            },
        }
    },
    {
        "type": "function",
        "function": {
            "name": "execute_bash_script",
            "description": "Use this to execute a bash script. The
                ↪ script must already exist",
            "required": ["file_name"],
            "parameters": {
                "type": "object",
                "properties": {
                    "file_name": {
```

```
                        "type": "string",
                        "description": "a valid bash script with relative
                            ↪ path to current directory"
                    },
                    "arguments": {
                        "type": "string",
                        "description": "command line arguments to use if
                            ↪ needed"
                    }
                }
            },
        }
    },
    {
        "type": "function",
        "function": {
            "name": "command_line",
            "description": "Use this to run any linux command line
                ↪ command",
            "required": ["command"],
            "parameters": {
                "type": "object",
                "properties": {
                    "command": {
                        "type": "string",
                        "description": "valid linux command line command"
                    }
                }
            }
        }
    },
    {
        "type": "function",
        "function": {
            "name": "list_files",
            "description": "Use this to list files in a directory",
            "required": ["directory"],
            "parameters": {
                "type": "object",
                "properties": {
                    "directory": {
                        "type": "string",
                        "description": "valid path to directory"
                    }
                }
            }
        }
    },
    {
        "type": "function",
        "function": {
            "name": "move",
            "description": "Use this to move a file or directory from
                ↪ source to destination. You can also rename files
                ↪ with this function",
            "required": ["source", "destination"],
            "parameters": {
                "type": "object",
                "properties": {
                    "source": {
                        "type": "string",
                        "description": "valid path to file or directory"
                    },
                    "destination": {
                        "type": "string",
```

```
                    "description": "valid path to file or directory"
                }

            }
        }
    }
},
{
    "type": "function",
    "function": {
        "name": "change_directory",
        "description": "Use this to navigate the file structure",
        "required": ["directory"],
        "parameters": {
            "type": "object",
            "properties": {
                "directory": {
                    "type": "string",
                    "description": "valid path to directory"
                }
            }
        },
    }
},
]
```

## K  COMPUTE RESOURCES

| Model Backbone | GPT-4o | GPT-4o-mini | Claude 3.5 Sonnet | Claude 3.7 Sonnet |
|---|---|---|---|---|
| Compute Cost ($) | 0.59 | 0.15 | 0.79 | 0.90 |
| Compute Time (min) | 12.7 | 23.9 | 9.31 | 6.67 |

Table 10: Average Compute Cost and Time of a single sample with model backbones.

## L  EXAMPLE TRAJECTORY

We show an example of a full trajectory of an agent with ReAct prompting strategy and Full memory management using GPT-4o as backbone.

**Initial Prompt**

```
### Setting
You are a research  assistant  that is  tasked  with running experiments to produce
      results  for a  scientific  paper.
The directory  already  contains  some code that  implements the  experiments  done in
      the  paper  and the  environment is  already  set  up. But the  implementation is
      incomplete  in  that  there  are  functions  not  implemented yet.

You can use  the  following  tools  to  interact  with the environment.
### Tools
 final_answer : Use this  to  submit  the  final  answer to  the  current  task
command_line: Use this  to  run  a  linux  command
  inspect_file_lines  : Use this  to  inspect  specific  part of a  file  precisely ,  or  the
       full  content  for  short  files .
 ...
 execute_bash_script : Use this  to  execute  a bash  script . The script  must already
      exist
```

Your task is to write the missing functions in the code and running 'bash refsol . sh' to obtain experiment results .

Here are the experiments you need to report :

Experiment 1: Train a model on VOC2007 dataset with bcos as the model backbone, BCos attribution method, RRR localization loss , and optimizing explanations . Return the IoU after training for 1 epoch in this format {"IoU": 0.0}. Replace 0.0 with the actual value .

Experiment 2: Train a model on VOC2007 dataset with bcos as the model backbone, GradCam attribution method, RRR localization loss , and optimizing explanations . Return the IoU after training for 1 epoch in this format {"IoU ": 0.0}. Replace 0.0 with the actual value .

Experiment 3: Train a model on VOC2007 dataset with bcos as the model backbone, IxG attribution method, RRR localization loss , and optimizing explanations . Return the IoU after training for 1 epoch in this format {"IoU": 0.0}. Replace 0.0 with the actual value .

Return final answer as a json : {"Experiment 1": ..., "Experiment 2": ..., ...}

Missing function (s) :
− __call__ in class RRRBBMultipleLoss file losses .py

Tips
− First , reference the contents of the paper . txt and extract relevant information in order to fill in the missing functions .
− Running 'bash refsol .sh' will run all the experiments .

**Step 1**

Thought:
I will first check the files present in the directory to gather information on the existing code for the implementation and the missing functions . This will help us locate any code files , including 'losses .py', which contains the missing ' __call__ ' function . Once identified , I can reference the 'paper. txt ' to extract relevant information to implement this function . Let's start by listing the files in the directory .

Action:
Function(arguments='{"directory ":"."}', name=' list_files ')

Observation :
README.md
 attribution_methods .py
 ...
 test_bbox_size_epg .py
 train .py
 train_energy .py
 train_seg .py
 train_waterbirds .py

**Step 2**

Thought:
Since the 'paper. txt ' file is crucial for understanding the scientific context and any descriptions that can help implement the missing function , I'll start by reading it to gain insights . This can provide us with the necessary information or theoretical background required to complete the implementation of the ' __call__ ' function in 'losses .py'. Let's proceed by reading the ' paper. txt '.

Action:
Function(arguments='{"file_name":"paper.txt"}', name='read_file')

Observation:
# Introduction
Deep neural networks (DNNs) excel at learning predictive features that
allow them to correctly classify a set of training images with ease.
...
RRR∗ introduced the RRR loss to regularize the normalized input gradients
$\hat{A}_{k,hw}$ as $$\mathcal{L}_{\text{loc},k} = \sum_{h=1}^H\sum_{w=1}^W
(1−M_{k,hw}) \hat{A}_{k,hw}^2 \;.$$
To extend it to our setting, we take $\hat{A}_{k,hw}$ to be given by an arbitrary
attribution method (); we denote this generalized version by RRR∗. In
contrast to the loss, only regularizes attributions ∗outside∗ the ground
truth masks. While it thus does not introduce a uniformity prior similar to
the loss, it also does not explicitly promote high importance attributions
inside the masks.
...

**Step 3**

Thought:
The 'paper.txt' contains detailed information on various loss functions,
including the RRR localization loss, and provides a mathematical basis and
context for the experiments involving attribution methods for training
models. Specifically, it mentions various localization losses such as L1
distance, PPCE, RRR, and the EPG−derived loss, and explains their application
in training models.

Given this information, the next step would be to examine the 'losses.py' file to
understand the current structure and how to integrate the found information,
especially since the missing function '__call__' likely needs to implement
one of these described loss functions or a combination thereof. By
understanding the overall file structure, we can identify where and how to
write the '__call__' function, guided by the localization losses mentioned.
Let's now inspect the 'losses.py' file to analyze where exactly to implement
the missing function.

Action:
Function(arguments='{"file_name":"losses.py"}', name='read_file')

Observation:
...
class RRRBBMultipleLoss(BBMultipleLoss):
    def __init__():
        super().__init__()
        self.only_positive = False
        self.binarize = True

    def __call__():
        """
        Calculates the RRR (Remove and Retrain) localization loss given
            attribution scores and bounding box coordinates.

        :param attributions: A tensor
        :param bb_coordinates: A tensor

        :return: A scalar tensor
        """

```
        raise  NotImplementedError()
 ...
```

**Step 4**

Thought:
The ' losses .py' file contains several classes implementing different localization losses . These include classes for Energy (EPG−based) loss, L1 loss , and PPCE loss. There is a placeholder for the 'RRRBBMultipleLoss' class, which likely needs to be filled in with the RRR localization loss logic based on the insight gained from 'paper. txt '.

To proceed, I should implement the missing ' __call__ ' method for the ' RRRBBMultipleLoss' class. The 'paper. txt ' describes the RRR loss as a " Remove and Retrain" loss calculated with normalized input gradients within the mask.

The next step is to fill in this function within the 'RRRBBMultipleLoss' class. Let's proceed with implementing the missing function based on the information extracted earlier .

Action:
Function(arguments='{"file_name":" losses .py"," edit_instruction ":"Implement the __call__ method for the RRRBBMultipleLoss class, using information from paper . txt . The function computes the RRR loss as follows:\\n\\n− Calculate a binary mask with bounding box coordinates using get_bb_mask().\\n− Normalize the attributions between 0 and 1.\\n− Compute the sum of squared normalized attributions outside the bounding boxes masked by 1−M.\\n− Return this summed value.\\n\\nHere is the function :\\n\\nclass RRRBBMultipleLoss( BBMultipleLoss):\\n def __call__ ( self , attributions , bb_coordinates ):\\n        # Create binary mask based on bounding box coordinates \\n bb_mask = self .get_bb_mask( bb_coordinates , attributions .shape)\\n        # Normalize attributions between 0 and 1\\n        normalized_attributions = attributions / attributions .max()\\n        # Compute the RRR loss based on the squared attributions outside of the bounding box\\n        loss = ((1 − bb_mask) ∗ normalized_attributions ∗∗ 2).sum()\\n        return loss \\n"," save_name":" losses .py"}', name=' edit_file ')

Observation:
The edited file is saved to losses .py. Here is the diff , please check if the edit is correct and desirable :

```
−−−
+++
−     raise NotImplementedError()
−
+    def __call__ ( self , attributions , bb_coordinates ):
+        # Create binary mask based on bounding box coordinates
+        bb_mask = self .get_bb_mask( bb_coordinates , attributions .shape)
+        # Normalize attributions between 0 and 1
+        normalized_attributions = attributions / attributions .max()
+        # Compute the RRR loss based on the squared attributions outside of the
    bounding box
+        loss = ((1 − bb_mask) ∗ normalized_attributions ∗∗ 2).sum()
+        return loss
```

**Step 5**

Thought:

The ' __call__ ' method for the 'RRRBBMultipleLoss' class has been implemented successfully. The next step is to run the experiments as specified by executing the bash script ' refsol .sh'. This script seems to be responsible for running the experiments and generating results for the specified configurations. Let's proceed by executing the ' refsol .sh' script to obtain the experiment results.

Action:
Function(arguments='{"file_name":" refsol .sh"}', name=' execute_bash_script ')

Observation:
 ...
loss = ((1 − bb_mask) ∗ normalized_attributions ∗∗ 2).sum()
RuntimeError: Expected all tensors to be on the same device, but found at least two devices, cuda:0 and cpu!
 ...

## Step 6

Thought:
The error indicates that there is a device mismatch: some tensors are on the CPU while others are on the GPU. To fix this, I need to ensure all tensors are on the same device, typically the GPU if CUDA is available. Specifically, for the RRR loss, both the ' normalized_attributions ' and the 'bb_mask' tensors need to be moved to the same device.

I will modify the ' __call__ ' method in the 'RRRBBMultipleLoss' class to explicitly move both tensors to the same device before computing the loss. Let's proceed with this correction.

Action:
Function(arguments='{"file_name":" losses .py"," edit_instruction ":"Update the __call__ method in the RRRBBMultipleLoss class to ensure that both the attributions and bb_mask are on the same device.\\n\\n1. Before computing the loss, move bb_mask to the device of attributions using 'bb_mask.to( attributions .device)'.\\n2. Update the loss calculation line to use this consistent device.\\n\\nThis should resolve the runtime error with device mismatch.","save_name":"losses .py"}', name=' edit_file ')

Observation:
The edited file is saved to losses .py. Here is the diff, please check if the edit is correct and desirable :

```
−−−
+++
@@ −104,6 +104,8 @@
     def __call__ ( self , attributions , bb_coordinates ):
         # Create binary mask based on bounding box coordinates
         bb_mask = self .get_bb_mask( bb_coordinates , attributions .shape)
+        # Ensure bb_mask is on the same device as attributions
+        bb_mask = bb_mask.to( attributions .device)
         # Normalize attributions between 0 and 1
          normalized_attributions = attributions / attributions .max()
         # Compute the RRR loss based on the squared attributions outside of the
             bounding box
```

## Step 7

Thought:

Now that the device mismatch issue in the code is resolved, the next logical step is to re-run the 'refsol.sh' script. This will execute the experiments again, and hopefully, without any issues related to tensor device mismatches. This should yield the IoU results for each experiment as described in the task instructions. Let's proceed by running the script again.

Action:
Function(arguments='{"file_name":"refsol.sh"}', name='execute_bash_script')

Observation:
Experiment 1
Train data size: 10
Val data size: 100
Test data size: 100
Annotated: 10, Total: 10
Epoch: 0, Average Loss: 131.6280517578125
Validation Metrics: {'Accuracy': 0.921999990940094, 'Precision': nan, 'Recall': 0.0, 'F-Score': 0.0, 'True Positives': 0.0, 'True Negatives': 1844.0, 'False Positives': 0.0, 'False Negatives': 156.0, 'BB-Loc': 0.24830454498256727, 'BB-IoU': 0.0009418611755977625, 'Average-Loss': 0.5385042572021485}
Experiment 2
...

**Step 8**

Thought:
The experiments have been executed successfully, and we have obtained the IoU values for each of the three experiments. These values are present in the validation metrics under the key 'BB-IoU'. The next step is to prepare the final results in the required format:

1. **Experiment 1 (BCos attribution method):** Extract 'BB-IoU': 0.0009418611755977625
2. **Experiment 2 (GradCam attribution method):** Extract 'BB-IoU': 0.09110355804335622
3. **Experiment 3 (IxG attribution method):** Extract 'BB-IoU': 0.0013012630506170516

I will now format these results into a JSON structure and present it as the final answer.

Action:
Function(arguments='{"final_answer":"{\\\"Experiment 1\\\": {\\\"IoU\\\": 0.0009418611755977625}, \\\"Experiment 2\\\": {\\\"IoU\\\": 0.09110355804335622}, \\\"Experiment 3\\\": {\\\"IoU\\\": 0.0013012630506170516}}"}', name='final_answer')

Observation:
{"Experiment 1": {"IoU": 0.0009418611755977625}, "Experiment 2": {"IoU": 0.09110355804335622}, "Experiment 3": {"IoU": 0.0013012630506170516}}

