# OpenReview forum: "From Reproduction to Replication: Evaluating Research Agents with Progressive Code Masking"
_ICLR.cc/2026/Conference — ICLR 2026 Poster_

### Official Review · Reviewer_3pHq · 2025-10-31

**Soundness:** 3
**Presentation:** 3
**Contribution:** 2
**Rating:** 4
**Confidence:** 4

**Summary:**

This paper presents AutoExperiment, a benchmark to evaluate LLMs and agents’ abilities to complete some functions in codebases that are necessary to replicate ML research papers’ results. With 85 functions adapted from four papers, the authors conduct controlled experiments by varying the number of functions to be implemented. Results show that agent performance drops as more functions are masked for implementation.

**Strengths:**

1. The proposed benchmark grounds its tasks on replicating real-world machine learning research papers.
2. The authors conduct a series of informative experiments, including controlled evaluation by masking different numbers of functions, exploring potential test-time scaling with verifiers, and trying different agent scaffolds.
3. The paper is clearly presented and easy to follow.

**Weaknesses:**

1. This paper proposes function masking and implementation as a new evaluation setting. However, it is questionable whether this is indeed necessary compared to related benchmarks (SciCode, ScienceAgentBench, PaperBench) that require LLMs to generate programs from scratch? Function completion essentially makes these research coding tasks less open-ended and easier by nature, which may fail to reflect real-world research coding.
2. As a benchmark, this work only involves 85 functions from four papers, which is too narrow and can be biased for evaluating LLMs and agents.

**Questions:**

Please see weaknesses.

---

> ### Author Response · Authors · 2025-11-20
>
> We would like to thank the reviewer for their insightful comments and feedback. In particular, we would like to highlight again the positive points mentioned in the review. We agree that the real-world applicative nature of the benchmark’s task is a strength of our paper. Furthermore, we appreciate that the reviewer found our analysis and ablations informative as we focused on crafting experiments to show the interesting aspects of our task and guide future research directions.
>
> Now, we would like to address some of the weaknesses:
> - **Function-masking setting vs from-scratch replication**: This is a tradeoff that we considered. Function completion is an easier task and less open-ended, but is easier to verify correctness. On the other hand, from-scratch replication (PaperBench, ScienceAgentBench) is harder and closer to real-world research coding, but is difficult to evaluate. To address this, PaperBench uses manually created “rubrics” to evaluate the code, but we find that this method is relatively subjective and less scalable in that the rubrics need to be manually created by human researchers. As a result, our benchmark uses function-level completion to make evaluation more concrete while allowing the task to scale in difficulty to almost real-world research coding by scaling the number of functions removed. Furthermore, we believe that function masking is a realistic setting for research work flows. Code infilling is the most common use case for AI copilots [1]. According to a human study on coding agent usage [2], using coding agents to generate new programs from scratch only takes up 25% while the rest of the use cases modify existing code including bug fixing and adding new features. Especially in research, it is important for human researchers to have granular and precise control over the implementation, which makes code infilling a more realistic task than from-scratch coding.
> - **Benchmark size**: This is a valid concern. We consider this as the first iteration of the benchmark, which includes high quality samples from a more manual process. In the future, we can incorporate more automatic pipelines (even using LLMs) to scale the dataset (a common scaling scheme in benchmarks like HumanEval [3, 4]). This version can act as the gold set to guide the generation of the larger silver set. We believe the strength of our paper lies in the benchmark creation framework, which can be applied to create future versions of the dataset to combat data contamination as well as scaling issues.
>
>
> [1] Chi, Wayne, et al. "Copilot arena: A platform for code llm evaluation in the wild." arXiv preprint arXiv:2502.09328 (2025).
>
> [2] Chen, Valerie, et al. "How can we assess human-agent interactions? Case studies in software agent design." arXiv preprint arXiv:2510.09801 (2025).
>
> [3] Chen, Mark. "Evaluating large language models trained on code." arXiv preprint arXiv:2107.03374 (2021).
>
> [4] Peng, Qiwei, Yekun Chai, and Xuhong Li. "Humaneval-xl: A multilingual code generation benchmark for cross-lingual natural language generalization." arXiv preprint arXiv:2402.16694 (2024).

---

> > ### Comment · Reviewer_3pHq · 2025-11-24
> >
> > Thanks for the responses! The first point makes sense, but I think my second concern wrt benchmark size is still substantial and not fully resolved by the promise of future (since we are discussing this version). Still, I've raised my score to weak accept.

---

### Official Review · Reviewer_4DjD · 2025-10-31

**Soundness:** 4
**Presentation:** 4
**Contribution:** 3
**Rating:** 6
**Confidence:** 4

**Summary:**

The paper introduces "AutoExperiment", a benchmark for evaluating AI agents for conducting scientific replication tasks. The benchmark consists of 85 functions that AI agents must "fill in" across 4 machine learning papers. The ML papers are drawn from MLRC. The benchmark is evaluated using ReAct (though other strategies are tried and work more poorly) across a number of closed and open-weight AI models. The authors draw compelling insights from this analysis.

Overall, the ability of AI to reproduce scientific work is an important research problem. The authors do a good job of constructing the benchmark, running the evaluations, and several ablations.

My main concern is about the small size of the benchmark. I appreciate that even with 85 missing functions, the benchmark is computationally expensive to run. But 4 papers seems like too small a sample size to conduct a benchmarking exercise. I would be more excited about a benchmark with 10+ papers, even if the number of missing functions per paper is smaller.

**Strengths:**

- ***Solid compute setup***: The computational setup for running the benchmark is decent, and well beyond what many other papers on scientific reproduction using AI agents use.
- ***Focus on reproducible results (such as by using Docker)***: I appreciate the authors' efforts to make their study reproducible, including using Docker, and making all of their code and analysis openly available.
- ***Qualitative analysis***: Many points in the paper include some qualitative analysis, which is quite insightful. For example, the authors analyze whether the first try passes or fails to explain what causes the difference between the fixed and dynamic settings.
- ***Nice ablation***: Many decisions taken in the paper are supported by strong experimental evidence. For example, the decision to run experiments with varying amounts of context about the paper (No vs. Full) is nice. (Are there intermediate versions of this you could run? Just providing the abstract/intro of the paper? Also, does this raise concerns about leakage, since instead of solving the paper's tasks, the agent could simply "answer" the questions based on the results in the paper?)

**Weaknesses:**

- In the introduction, there's a typo ("agentive", should this be "agentic"?)
- No train-test split: As far as I understand, there's no "train" split in the data. How should researchers using the benchmark optimize and develop their agents without a train set? This raises concerns about leakage (even in the results reported in the paper, since there are many ablations, for example, in the ReAct agents)
- No reporting of cost, time taken by agents (which can be important for real-world use)
- Why pick these 4 papers? MLRC has dozens of papers from the last few years, is there a reason these 4 specific papers were picked?
- It would be nice to see pass^k in addition to pass@k: Papers like TauBench report pass^k (whether all k samples correctly solved the task). This could be interesting to see the reliability of the agents.
- The verifier used in the paper is quite simple. The authors could test with better verifiers to see if pass@k performance can be distilled to pass@1 (though it's fine to leave this for future work).
- How did the authors decide on the 5% tolerance for declaring a test case correct? Did you use the human replication to come up with this tolerance level?
- It's unclear where the curtailed tests are used and where the full tests are used
- It's unclear if all models are evaluated on the same 100 examples for n>1, or whether these are randomly sampled each time. If the latter, we might need CIs for more reliable results.

**Questions:**

Overall, I think this paper is a decent contribution. However, the small sample size, the lack of a test set, and lack of clarity about my questions above leads me to propose a borderline score for the paper. That said, I would be happy to increase my score if the following points are addressed:

1) The authors commit to adding a held-out test set (perhaps with 4 other MLRC papers). Of course, I realize the authors can't run experiments on this test set. That said, it would be nice to have a held-out set and functions that go beyond the ones that agent developers might use for "training"/developing their agents. It would also make the paper's results more reliable, since n=4 is a very small sample size.

2) The authors address the weaknesses I mention above, in particular to improve the clarity of the paper. (I understand if some of these weaknesses are hard to address, such as reporting cost and time taken after the fact, but I hope the others are easier to address.)

---

> ### Author Response · Authors · 2025-11-20
>
> Thank you for a detailed and insightful review. In particular, we appreciate the reviewer’s comments on the strength of our benchmark’s evaluation infrastructure as well as insightfulness of our analysis and ablation experiments.
>
> - **Train-Test split**: Many recent coding/agentic benchmarks are also test-only benchmarks such as PaperBench [1] and SUPER [2]. Furthermore, it has become common practice for follow-up work to create their own training sets. For instance, follow-up works to SWE-Bench [3] like SWE-RL [4] and SWE-Smith [5] have created their own train sets by adapting the data collection process from SWE-Bench. Similarly, our benchmark’s current dataset can be used as a high-quality test-set and the same data collection process can be used to create larger train sets. Since the goal of our paper was to introduce progressive masking as a benchmark paradigm for scientific and agentic coding, we believe creating a train set is more suitable for future work.
> - **Cost and time**: As described in the “Environment and Resources” section in Section 2, we included the average cost and time of an experiment in Appendix K.
> - **Paper selection**: We manually selected papers that were of high-quality and easily replicable. We found that many other papers had code repositories that were difficult to create working environments for or their code did not run properly without making substantial edits to the existing codebase. Hence, we filtered top repositories that had well-managed documentation and working Conda environments.
> - **Pass^k metric**: Thank you for the suggestion! Interestingly, we found that the Pass^5 rate for GPT-4o is 9.4% and Claude-3.5-Sonnet is 18.8%. This result helps to validate our previous Pass@1 vs Pass@5 comparison. The gap in performance is much higher for GPT-4o than Claude-3.5-Sonnet, suggesting that GPT-4o has more variance in its responses and chosen actions. We will be including this metric in Figure 3 in the final version of the paper.
> - **Verifier**: We wanted to keep the verifier to a simple version to demonstrate the pass@k gap and encourage future work in this direction.
> - **Tolerance Level**: The error tolerance was motivated by prior related work (SUPER [2]), which tests agents' ability to run scripts given a full codebase. Although SUPER does not require agents to write code, the evaluation scheme is very similar to ours.
> - **Curtailed experiments**: Sorry about the confusion! During the dataset creation process, we create curtailed versions of tests when the time to run the tests are too long. All experimental results from the paper are using these curtailed tests.
> - **Evaluation samples**: All models and agent architectures are evaluated on the same 100 examples for fairness.
>
> [1] Starace, Giulio, et al. "PaperBench: Evaluating AI's Ability to Replicate AI Research." arXiv preprint arXiv:2504.01848 (2025).
>
> [2] Bogin, Ben, et al. "Super: Evaluating agents on setting up and executing tasks from research repositories." arXiv preprint arXiv:2409.07440 (2024).
>
> [3] Jimenez, Carlos E., et al. "Swe-bench: Can language models resolve real-world github issues?." arXiv preprint arXiv:2310.06770 (2023).
>
> [4] Wei, Yuxiang, et al. "Swe-rl: Advancing llm reasoning via reinforcement learning on open software evolution." arXiv preprint arXiv:2502.18449 (2025).
>
> [5] Yang, John, et al. "Swe-smith: Scaling data for software engineering agents." arXiv preprint arXiv:2504.21798 (2025).

---

### Official Review · Reviewer_VcpW · 2025-10-31

**Soundness:** 3
**Presentation:** 3
**Contribution:** 2
**Rating:** 6
**Confidence:** 3

**Summary:**

The authors propose a new benchmark, AutoExperiment, that tests if AI agents can implement ML experiments described in research papers. The benchmark is based on masking out functions for the agent to fill in (scalable by # functions masked). They evaluate various agents and find performance degrades with # fns masked, and that self-reflection helps.

**Strengths:**

- Appropriate and opportune benchmark - scientific coding is rapidly becoming a major research area, and we still lack good benchmarks
 - Clear and well written
 - Builds on prior work (MLRC) nicely
 - Nicely scalable levels of difficulty in the benchmark

**Weaknesses:**

- Most significantly, this paper seems very similar to "ResearchCodeBench: Benchmarking LLMs on Implementing Novel Machine Learning Research Code" (https://arxiv.org/abs/2506.02314), significantly weakening the paper's claims of novelty. Please describe the key differences/benefits of your work to this prior paper.
 - The stated findings from the experiments seem obvious (e.g., performance degrades with more masking; debugging helps). What surprising/discoveries did you make in your experiments? What did you learn that was not obvious before? The most significant take-away to me is that agents still struggle with this kind of coding (e.g., most scores are < 50%) and more work needs to be done.

**Questions:**

See Weaknesses. Also:
 - In what way is "scientific" coding different to general coding, or is it the same thing? Are there some specific characteristics of coding that your benchmark captures that other coding benchmarks do not?
 - Is it always a clear, single gold answer to the experiment? In particular, NL descriptions of an experiment are typically underspecified, meaning there might be several ways of implementing an experiment with different design decisions (e.g., choice of hyperparameters), and correspondingly different results.
 - Filling in missing functions seems a somewhat artificial context - in reality, the agent simply has to code up an experiment. Do you think scores on filling in missing functions correlate with scores on coding up an experiment from scratch (the metric we ultimately care about)? Or do your experiments substantiate that correlation?
 - Your scores on AutoExperiment show models still struggle (~< 50%), but other coding benchmark tests have shown agents are strong at coding (e.g., HumanEval, SWE-Bench). How do you explain this apparent inconsistency?

Minor:
 - Expand the MLRC acronym (Section 2) at first mention, not later.
 - I'd prefer the Related Work earlier (Section 2) rather than an afterthought at the end, given the considerable related work in this area

---

> ### Author Response · Authors · 2025-11-20
> **Response to Weaknesses**
>
> Thank you for a detailed review and constructive feedback! First, we would like to echo the strengths of our paper: the benchmark tests LLMs’ ability in scientific coding, an important task that there does not exist a strong benchmark for. We also agree with the reviewer that the benchmark’s unique contribution is its ability to control the difficulty in a granular and scalable way. Now on to the constructive feedback:
>
> **Weaknesses**:
> - **Comparison to ResearchCodeBench**: ResearchCodeBench [1] is a recent benchmark, which has been published on Arxiv and accepted to the upcoming Neurips conference. According to ICLR policy), this work is considered contemporaneous since the paper is not published in a peer-reviewed journal at least 2 months prior to ICLR submission deadline. ResearchCodeBench is similar to our benchmark in that they test function-level code infilling in research code repositories, evaluate based on execution results, and analyze results across different natural language contexts. However, one key difference is that their tasks are defined manually and they are fixed for each paper while our benchmark can scale the difficulty via progressive masking. Furthermore, our benchmark is inherently ‘agentic’ as demonstrated in section 3.3. In the context of scientific coding, our benchmark tests realistic, agentic coding tasks that can scale to full replication (analogous to SWE-Bench) while ResearchCodeBench tests LLM code-infilling abilities (analogous to HumanEval). Thus, we believe our benchmark offers a more future-aware way to evaluate complex coding agents. We will be citing ResearchCodeBench and including the comparisons in the final version of the paper.
> - **Key takeaways**: Thank you for mentioning the key findings of the paper. We agree that a main discovery is that agents struggle with scientific coding especially as the amount of masked code increases. Another key finding is that this task requires agentic interactions with the environment rather than scaling reasoning tokens. This lays the foundation for future work to focus on developing agents that take advantage of agent interactions. Furthermore, we found that there is a large pass@k gap, which motivates research in developing verifiers/rerankers that can boost pass@1 performance or training the agent to become more reliable research coders. Lastly, we found it interesting that the reliance on natural language to write code becomes stronger as we mask more functions in section 3.4. This highlights the importance of context retrieval in agentic frameworks as tasks become more complex.
>
> [1] Hua, Tianyu, et al. "ResearchCodeBench: Benchmarking LLMs on Implementing Novel Machine Learning Research Code." arXiv preprint arXiv:2506.02314 (2025).

---

> > ### Author Response · Authors · 2025-11-20
> > **Response to Questions**
> >
> > **Questions**:
> > - **Scientific coding**: In general, scientific coding requires implementing novel ideas in specific subareas of science. In our case, we focus on machine learning research coding, which involves analyzing task-specific data, implementing new methods or model architectures, and coding with frameworks like PyTorch and Tensorflow.
> > - **Ambiguity in gold result**: This was a concern that we faced during benchmark construction and we have taken steps to make our task as unambiguous as possible. Our “testcases” are scripts that run the implemented code with specifically defined hyperparameters, which removes much of the ambiguity. By testing implementation at a function level, we hope to make this natural language to code mapping as concrete as possible. We believe that this is a key distinguishing feature of our benchmark compared to full replication benchmarks like PaperBench, which have to rely on natural language rubrics for evaluation.
> > - **Code-infilling setting**: The benefit of our benchmark is that we can scale up $n$ (number of functions masked out), which converges to the task of coding up an experiment from scratch. We see masked code-filling as a subtask of the overall implementation-from-scratch task. For instance, when writing code, a function header is often written first and the implementation is left to be filled out later. Furthermore, studies have found that code infilling is the most common use case for AI copilots [2]. According to a study of how developers use coding agents [3], using coding agents to generate new programs from scratch only takes up 25% of all usages while the rest of the use cases modify existing code including bug fixing and adding new features. Especially in research, it is important for human researchers to have granular and precise control over the implementation, which makes code infilling as realistic as from-scratch coding with very high correlation.
> > - **Consistency vs other coding benchmarks**: Compared to HumanEval, our task is inherently more difficult since it requires understanding of research-level NL ideas and navigating and searching from a repository. In terms of difficulty, our task is closer to SWE-Bench, which is also repository level and requires understanding of natural language Github issues. For SWE-Bench, the current leaderboard suggests that the best agent achieves around 78% in the Verified set (https://www.swebench.com/). Keeping in mind that researchers have been working on SWE-Bench for a while, our benchmark results are not necessarily inconsistent with SWE-Bench. Also, the fact that our benchmark is more difficult than SWE-Bench allows for more headroom to improve agents in future work.
> >
> > [2] Chi, Wayne, et al. "Copilot arena: A platform for code llm evaluation in the wild." arXiv preprint arXiv:2502.09328 (2025).
> >
> > [3] Chen, Valerie, et al. "How can we assess human-agent interactions? Case studies in software agent design." arXiv preprint arXiv:2510.09801 (2025).

---

> > ### Comment · Reviewer_VcpW · 2025-11-25
> >
> > Thank you for the clarifications - indeed ResearchCodeBench is contemporaneous, however it would be good to mention it. Your points of comparison, in particular the scalability of AutoExperiment, are nice and worth contrasting with ResearchCodeBench. Also thank you for clarifying the takeaways.

---

### Official Review · Reviewer_i5wK · 2025-11-01

**Soundness:** 2
**Presentation:** 2
**Contribution:** 2
**Rating:** 4
**Confidence:** 4

**Summary:**

This paper propose a new benchmark——AUTOEXPERIMENT，for evaluating the ability of agents to implement ml experiments. Tasks are created by masking out various numbers of functions from ML research repo codebases and requiring agents to fill in the blanks and match published results. The paper also evaluates several LLMs, presenting detailed analyses of different aspect.

**Strengths:**

1. The paper proposes a new benchmark for evaluating the performance of agents on machine learning code replication tasks.
2. It evaluates the performance of different models and agents under varying task settings (i.e., partial code replication and full replication).

**Weaknesses:**

1. Novelty of the Benchmark: The benchmark is presented in two settings: partial code replication and full code replication. For full replication, a significant body of related research already exists. The partial replication setting closely resembles many existing code-filling tasks studied in the context of code models.
2. Rigorousness of Experimental Setup: The paper uses the number of masked functions (n) as its primary analysis target and difficulty metric, concluding that a larger 'n' decreases the likelihood of successful replication. This conclusion is trivial and has already been well-established in numerous code-filling tasks. In practice, different functions vary dramatically in logical complexity, code length, and library dependencies. Using 'n' as a universal measure of difficulty is overly simplistic.
3. Lack of Diverse Experimental Subjects: The paper's main results are derived from models like GPT-4o and Claude-3.7, which have known capability limitations. As of 2025, many more powerful reasoning models have emerged that demonstrate superior performance on software engineering tasks, but this paper fails to evaluate them. Moreover, the agent setup itself is rudimentary, employing only a simple tool-using agent designed by the authors. This setup does not reflect the current state-of-the-art in agent capabilities and appears to be testing the backbone model more than the agent's autonomous abilities.
4. Reasonableness of the Benchmark Design: The paper states that with an increasing number of masked functions, the 85 samples from just 4 papers can lead to 275,990 possible combinations for n=5. The methodology of sampling only 100 of these combinations offers no guarantee that these samples are representative of that difficulty level. The design of this benchmark appears to be sloppy and lacks statistical rigor, as this sparse sampling introduces massive variance, making comparisons between different 'n' levels unreliable.

**Questions:**

See above.

---

> ### Author Response · Authors · 2025-11-20
>
> Thank you for a thoughtful review of our work!  We would like to address your constructive feedback.
>
> - **Novelty of the benchmark**: Full replication benchmarks like PaperBench [1] test AI agents’ ability to implement scientific code from scratch. This process has a large degree of freedom, leading benchmarks like PaperBench to use natural language rubric based evaluation methods. We believe this method is not as scalable as execution-based evaluation (ours) since rubrics need to be manually created and can be subjective to humans and LLM judges. For the partial code completion setting, our benchmark differs from other code-infilling benchmarks which mostly focus on general coding. Scientific coding (ours) differs from general coding in that it requires a deep understanding of a scientific field and implementation of novel research ideas. Most importantly, our work is unique in that we can control the knob of difficulty from reproduction to from-scratch replication while maintaining concrete execution-based evaluation. We believe the benchmark captures the unique challenges of scientific coding and can act as a good testbed to test AI agents for scientific discovery.
> - **Rigorousness of Experimental Setup**: We would like to split the feedback into 2 main concerns: a) the finding that performance decrease as ‘n’ increase is basic and already known. b) using ‘n’ as number of functions removed as measure of difficulty is too simplistic since functions differ in code length, library dependence, etc
>     - To address the first concern, we agree that it does seem obvious that performance decreases as we remove more functions. The motivation behind this experiment was to highlight that our benchmark can be controlled in terms of difficulty. Currently, the agents do reasonably well with 1 function removed, but performance falls quickly. As better agents/models develop, we can scale the difficulty of the benchmark, which allows us to guide the path towards full from-scratch replication. Furthermore, we believe that our subsequent experiments offer insightful findings such as the large pass@k gap, need for agent interaction, and dependence on natural language.
>     - To address the second concern, we agree that the “complexity” of a function can be measured in different ways such as code length, number of function calls, etc. Although using ‘n’ as a measure of difficulty is simplistic, our results show that performance drops quickly as a function of n. For now, we believe n is granular enough, but in the future, we believe that analyzing other factors such as code length and function calls is a promising direction to guide the development of better research agents.
> - **Lack of Diverse Experimental Subjects**:
>     - **New Models**: In section 3.3, we tested reasoning models like o1 and o3-mini. In particular, when using o3-mini as the backbone LLM in an agent, it did not perform better than gpt-4o (Table 3). To further address this concern, we ran a small experiment (with 50 samples) using the new GPT-5.1 as backbone LLM, and found that the performance follows similar trends. This agent achieves around 40% pass rate in n=1 setting, followed by 16% in n=2, and 8% in n=3.
>     - **Agent Scaffolds**: Our agent setup is based on SWE-agent [2], which still is one of the top performing scaffolds according to SWE-Bench leaderboard (https://www.swebench.com/). Although new agent scaffolds exist (OpenHands, TRAE-agent), the specifics of agent architecture are still unclear and under research (especially in the research coding domain). Hence, we focus our analysis on whether agentic interaction is needed (section 3.3) and perform simple ablations on agent architecture (Appendix C), which can act as a starting point for future agent research.
> - **Reasonableness of the Benchmark Design**: We agree that with the small sample size the comparison between performance when n >= 3 may not be significant. However, from n=1 to n=2, the performance drops significantly. With the current abilities of AI agents, we believe it will not be more insightful to use more samples for n >= 3 settings if the conclusion will be the same that agents struggle with this task. As agent performance improves in the future, our benchmark can be used to its full capacity (by sampling more) to more granularly examine models in n >= 3 settings.
>
> [1] Starace, Giulio, et al. "PaperBench: Evaluating AI's Ability to Replicate AI Research." arXiv preprint arXiv:2504.01848 (2025).
>
> [2] Yang, John, et al. "Swe-agent: Agent-computer interfaces enable automated software engineering." Advances in Neural Information Processing Systems 37 (2024): 50528-50652.

---

> > ### Comment · Reviewer_i5wK · 2025-11-28
> >
> > Thanks for the response. After reading the response, I'd like to raise the score to 6, if possible.

---

### Author Response · Authors · 2025-12-03
**Summary for AC**

We would like to take this opportunity to summarize the feedback received. Reviewers VcpW and 4DjD gave a score of 6 and while reviewers i5wK and 3pHq initially identified weaknesses, after our rebuttal, they raised their scores to 6 as well.

**Strengths**

The reviewers mentioned that our benchmark focuses on **evaluating AI agents on machine learning research replication**, an important rising area where we lack good benchmarks (reviewer i5wK, VcpW). We agree that our benchmark setting is crucial for future development of AI research agents in that it tests a **realistic code-infilling task** in the context of scientific coding. In addition, our benchmark provides a **controllable setting** in which we can **interpolate the difficulty of the task** from partial to full replication - allowing us to effectively test foundation and frontier models at growing levels of capability (reviewer VcpW). We agree that this is one of the core strengths of our benchmark since the granularity in difficulty provides a clear way to evaluate different models and agent architectures while guiding the path towards full autonomy. Furthermore, the reviewers took notice of our **insightful ablation experiments and key findings** (reviewer 4DjD, 3pHq). Our experimental results offer key insights such as large pass@k gap, which motivates verifier approaches (section 3.2). We also demonstrate the importance of agent interactions (3.3) and natural language dependence (3.4). These findings provide deeper insight into the nature of our task and offer clear directions for future work. Lastly, we greatly appreciate the reviewers’ comments about our effort to make our method as **reproducible** as possible via Docker containers, etc (reviewer 4DjD), and make our findings **clear and easy-to-follow** (reviewer VcpW, 3pHq).

**Weaknesses**

- **Benchmark setting (progressive code masking)**: Reviewer 3pHq initially had concerns about our code infilling setting. Comparing the setting to from scratch replication, we demonstrated that masked code-infilling is a subtask of the overall implementation-from-scratch task. Citing recent studies on AI coding agent and copilot usage [3, 4], we showed that the majority of use cases involve modifying code rather than writing code from scratch, especially for scientific code. As a result of this response, reviewer 3pHq raised their score to 6.
- **Novelty**: Some reviewers had concerns about the novelty of our benchmark setting (reviewer i5wK, VcpW). We compared our benchmark to full replication benchmarks like PaperBench [1], which differ in that they use rubric-based evaluation methods while we use a more concrete execution-based evaluation. We also compared our benchmark to ResearchCodeBench, a contemporaneous work according to ICLR policy. Our benchmark is unique in that our task is inherently agentic and we can easily scale the difficulty of the benchmark to full autonomy. Reviewer VcpW agreed with our comparison, which we will include in the final version of the paper. Also, reviewer i5wK raised their score to 6 after our response.
- **Additional experiments/metrics**: We ran additional experiments with newer models (reviewer i5wK) and found that similar trends hold (performance drops as n increases). We also computed the pass^k metric (reviewer 4DjD) and we found results that help validate our previous Pass@1 vs Pass@5 comparison.
- **Dataset/Sample size**: Reviewer i5wk expressed concern for the small sample size in n>=3 settings. We explained that as models get better at the task, we can increase the number of evaluation samples in n>=3 settings, providing more granular feedback. After our response explaining the novelty of the benchmark, additional experiments, and flexible sample size, reviewer i5wK raised their score to 6. Reviewer 3pHq mentioned the overall dataset size. We believe that the strength of the benchmark comes from a scalable data generation framework that could be loosened to create larger “silver” sets in the future.

In addition, we have answered all other minor questions about the paper in our rebuttal. For the final version of the paper, we plan to fully incorporate the feedback from reviewers by 1) adding more detail comparing our benchmark’s setting with prior work like PaperBench and ResearchCodeBench (contemporaneous work), 2) including pass^k metric in our figure 3 and analysis, and 3) adding clear instructions on increasing sample size for n>=3 settings.

---

> ### Author Response · Authors · 2025-12-03
> **Citations**
>
> [1] Starace, Giulio, et al. "PaperBench: Evaluating AI's Ability to Replicate AI Research." arXiv preprint arXiv:2504.01848 (2025).
>
> [2] Hua, Tianyu, et al. "ResearchCodeBench: Benchmarking LLMs on Implementing Novel Machine Learning Research Code." arXiv preprint arXiv:2506.02314 (2025).
>
> [3] Chi, Wayne, et al. "Copilot arena: A platform for code llm evaluation in the wild." arXiv preprint arXiv:2502.09328 (2025).
>
> [4] Chen, Valerie, et al. "How can we assess human-agent interactions? Case studies in software agent design." arXiv preprint arXiv:2510.09801 (2025).

---

### Meta-Review · Area_Chair_DPvy · 2026-01-05

**Summary:**

The main concern raised by multiple reviewers was the novelty of the benchmark. Reviewers questioned whether progressive code masking meaningfully differs from existing code-infilling or full-replication benchmarks such as PaperBench and ResearchCodeBench. In response, the authors clarified that AUTOEXPERIMENT uniquely provides a continuous difficulty spectrum between reproduction and full replication, while retaining execution-based evaluation rather than rubric- or LLM-judged scoring. They further emphasized that their benchmark is inherently agentic and supports systematic scaling toward full autonomy, which reviewers acknowledged after the rebuttal, leading to score increases.

Another recurring concern was the use of the number of masked functions (n) as a difficulty metric, which some reviewers viewed as overly simplistic and leading to obvious conclusions (performance decreases as n increases). The authors agreed that n is a coarse measure, but argued that its purpose is to act as a controllable difficulty knob rather than a fine-grained complexity metric. They highlighted that the non-trivial contributions lie in secondary findings, including the large Pass@k gap, the importance of agentic interaction over static pipelines, and the increasing reliance on natural language as code context diminishes.

Reviewers also questioned the statistical rigor of sampling, especially for larger n where only a small subset of possible combinations is evaluated. The authors responded that for current models, performance collapses sharply beyond n=2, making larger sample sizes less informative at present. They framed AUTOEXPERIMENT as a scalable benchmark framework whose evaluation granularity can increase as agent capabilities improve.

Concerns about the limited diversity and strength of evaluated agents were addressed by adding experiments with reasoning models and newer backbones, which showed similar performance trends. The authors justified their agent scaffold by grounding it in SWE-agent and clarified that the goal was to study the role of agentic interaction rather than propose a new agent architecture.

Finally, reviewers raised questions about the realism of scientific coding and ambiguity in gold results, noting that scientific implementations are often underspecified. The authors explained that ambiguity is reduced through function-level masking and fixed experimental scripts, and argued that masked infilling is a principled subtask that converges to full replication as more code is removed.

Overall, the rebuttal effectively clarified the benchmark’s positioning, strengthened empirical support, and addressed concerns about rigor and realism, leading several reviewers to raise their scores. Thus, I recommend acceptance.

**Reviewer Concerns:**

Most of the concerns have been addressed by authors' rebuttal.

**Reviewer Scores:**

Reviewers VcpW and 4DjD assigned scores of 6, and i5wK and 3pHq raised their scores to 6 after the rebuttal, which I agree with.

---

### Decision · Program_Chairs · 2026-01-26

Accept (Poster)